# Effect of Longitudinal Practice in Real and Virtual Environments on Motor Performance, Physical Activity and Enjoyment in People with Autism Spectrum Disorder: A Prospective Randomized Crossover Controlled Trial

**DOI:** 10.3390/ijerph192214668

**Published:** 2022-11-08

**Authors:** Íbis A. P. Moraes, Joyce A. Lima, Nadja M. Silva, Amanda O. Simcsik, Ana C. Silveira, Lilian D. C. Menezes, Luciano V. Araújo, Tânia B. Crocetta, Mariana C. Voos, James Tonks, Talita D. Silva, Helen Dawes, Carlos B. M. Monteiro

**Affiliations:** 1Rehabilitation Sciences, Faculty of Medicine, University of São Paulo (FMUSP), São Paulo 01246-903, Brazil; 2College of Medicine and Health, St Lukes Campus, University of Exeter, Exeter EX1 2LU, UK; 3Physical Activity Sciences, School of Arts, Science and Humanities, University of São Paulo (EACH-USP), São Paulo 03828-000, Brazil; 4Medicine (Cardiology), Escola Paulista de Medicina, Federal University of São Paulo (EPM/UNIFESP), São Paulo 04021-001, Brazil; 5Faculty of Humanities and Health Sciences, Pontifical Catholic University of São Paulo (PUC-SP), São Paulo 05014-901, Brazil; 6Faculty of Medicine, University City of São Paulo (UNICID), São Paulo 03071-000, Brazil; 7Department of Paediatrics, University of Oxford, Oxford OX3 9DU, UK

**Keywords:** Autism Spectrum Disorder, virtual reality, exercise, motor skills, heart rate

## Abstract

(1) Background: People with ASD commonly present difficulty performing motor skills and a decline in physical activity (PA) level and low enjoyment of PA. We aimed to evaluate whether longitudinal practice of an activity in virtual and real environments improves motor performance and whether this improvement is transferred to a subsequent practice when changing the environment, promoting PA and providing enjoyment; (2) Methods: People with ASD, aged between 10 and 16 years, were included and distributed randomly into two opposite sequences. The participants performed a 10 session protocol, with five sessions practicing in each environment (virtual or real). Heart rate measurement was carried out and an enjoyment scale was applied; (3) Results: 22 participants concluded the protocol. Sequence A (virtual first) presented an improvement in accuracy and precision and transferred this when changing environment; they also had a greater change in heart rate reserve. The majority of participants reported “fun” and “great fun” levels for enjoyment; (4) Conclusions: The virtual reality activity presented a higher level of difficulty, with greater gains in terms of transference to the real environment. Considering PA, our task provided very light to light activity and the majority of participants enjoyed the task.

## 1. Introduction

Autism Spectrum Disorders (ASD) are among the most common developmental disabilities [1]. Approximately 1 in 44 children have ASD according to estimates from the Center for Disease Control and Prevention associated with the Developmental Disabilities Monitoring Network [2]. ASD is characterized by a quantitative deficit in communication and social interaction, along with a series of restricted, repetitive and stereotyped behaviors and interests [3]. ASD is more frequently diagnosed in boys than girls, with the proportion 4–5:1 [2,4]. ASD has a large degree of interindividual variability with respect to both the severity and profile of symptoms [5]. This type of symptomatology, according to the DSM-5, is constituted by hyper- or hypo-reactivity to sensory input or unusual interest in sensory aspects of the environment [6]. In this sense, it has been suggested that multisensory integration is impacted in ASD [7,8]. Sensory abnormalities may negatively impact the life of these individuals and their families and, importantly, their sensorimotor functioning [9], generating the incapacity to properly learn internal models [10,11,12,13].

According to a recent systematic review, difficulty performing age-appropriate motor skills affects up to 83% of children with ASD [14]. In addition, these individuals show significative impairments in motor imitation, visual-motor function and hand motor activities [1,15,16]. Until now, the majority of treatments for people with ASD have focused on the communication and social ambit deficits, with only a few studies and interventions aiming to improve motor abilities [17]. Furthermore, people with ASD frequently show a decline in physical activity (PA) level, related to all the aforementioned deficits (social, behavioral and motor) [18], as well as to factors such as lack of motivation [19], low interest [20,21] and low enjoyment of PA [22]. Health promotion activities are currently focused on searching for tools that popularize modern physical activity, making it attractive for people [23]. Research studies showed that Virtual Reality (VR) has been used as a tool to fight hypokinesia, eliminating periods of lack of PA [24] and reducing negative sensations associated with PA [25]. The use of VR technology in training programs may be a factor that increases motivation for PA [26]. Furthermore, VR activities that are more enjoyable and engaging may avoid sedentary behavior in children and adolescents, which leads to cardiovascular diseases in adults [27,28].

The use of VR allows therapists to offer a secure platform that can be used to implement body movements, performed in a variety of interesting environments, with and without physical contact, to interact with the task, providing better learning of motor skills for people with ASD [29]. VR also raises the probability of these abilities being transferred to their own everyday lives [30,31,32], as demonstrated by Moraes et al. [33], who used a cross-sectional protocol and concluded that people with ASD were able to transfer improvements gained from practicing in a virtual environment to a real environment. To date, no studies have considered the transfer of motor abilities between virtual and real environments with a prolonged intervention. In the current study, we tested the hypothesis that the longitudinal practice of an activity in virtual and real environments would promote: (1) improvement in motor performance which would be transferred to subsequent practice when changing the environment, (2) physical activity by altering the heart rate, (3) enjoyment.

## 2. Materials and Methods

This study was approved by the research ethics committee of the University of São Paulo (CAAE: 79411217.3.0000.0065) and was registered in the Registro Brasileiro de Ensaios Clínicos (ReBEC) database, with the following identifying number: RBR-6c36pg. This is a longitudinal prospective randomized crossover controlled trial. This study was carried out at the Integrated Psycho-Pedagogical Support Group (GAPI)—Special Education school in São Bernardo do Campo, São Paulo, Brazil—an institution specialized in children and adolescents with developmental disorders. The study intervention period was between December 2017 and July 2019. This study follows the guidelines and recommendations of the Consolidated Standards of Reporting Trials 2010 guideline—CONSORT 10 [34,35].

### 2.1. Participants

Volunteers aged between 10-and-16 years old, with a diagnosis of ASD confirmed by a neurologist and by the multidisciplinary team through review of neuro-developmental history, psychological assessment, communication assessment and psychiatric assessment, participated in the study.

Our inclusion criteria required: (1) the signing of the assent term by the participant and the free and informed consent form by the parents or guardians of the minors; (2) children and adolescents with mild and moderate ASD, who could understand the tasks proposed; and (3) who did not use medication that could interfere with the studied variables, such as cardiac betablockers. As further exclusion criteria, we considered: (1) children without comorbidities such as Attention Deficit Disorder with Hyperactivity and Down Syndrome (2) children who were unable to understand the task, assessed through the ability of the participant to perform the task correctly after three supported attempts with explanations and demonstrations from the evaluator—1 exclusion (3) who withdrew during the study—1 exclusion.

To determine the sample size, we used statistical software (G∗Power 3.1.5) on the main outcome measure (i.e., the motor score). This calculation was based on data from five patients (pilot study). The power was 0.80, the α was 0.05 and the effect size was 0.65 (Cohen d). The sample estimation indicated that 20 participants would be necessary (i.e., 10 per group) and after an adjustment to allow for withdrawal rate (20%), we included 24 participants.

### 2.2. Sample Characterization

The initial assessment for sample classification was carried out two days before the start of the intervention. To evaluate the Intelligence Quotient (IQ), we used the abbreviated version of the Wechsler intelligence scale (WISC III), which was applied by a psychologist at the institution [36,37]. The IQ is classified into mild intellectual disability, with a score of 55–70, borderline intelligence with a score of 70–85, normal intelligence with a score of 85 or above, above average intelligence with a score of 115–129 and superior intelligence with a score of 130 or above [38]. The severity of autism was assessed by the Childhood Autism Rating Scale (CARS), classified according to a total score, with 15 to 29 considered as “non-autistic”, 30 to 36 as “mild to moderate” ASD and finally, 37 to 60 as “moderate to severe” ASD [39,40]. Lastly, the Pediatric Evaluation of Disability Inventory (PEDI) was applied to obtain a detailed description of the child’s functional performance. Capacity and performance are documented into two domains: 1. Functional Skills and 2. Caregiver assistance, divided according to three different areas: 1. Self-care, 2. Mobility and 3. Social Function [41,42].

### 2.3. Virtual Reality (VR) Activity

The participants were comfortably positioned in a chair that was adjusted according to their size and needs. Before starting the task, a verbal explanation and a demonstration on how to play the VR were given by the examiner. The researchers involved in the application of the intervention protocol did not participate in the recruitment, initial evaluation and randomization stages.

The game was developed by the School of Arts, Sciences and Humanities of the University of São Paulo (EACH/USP) and includes falling spheres, in four imaginary columns on the computer screen, to the rhythm of a song, chosen by the researcher, according to the confines of the software [43,44,45]. The task consists of not allowing the spheres to fall, but the spheres can only be grabbed when they reach their target (a total of four targets), which is the same color as the sphere and these targets are positioned in parallel (at two heights), two on the left and two on the right (Figure 1). This software is an improvement on a previous version that was used in other protocols focused on improving motor performance in ASD [33] and other developmental disabilities [46,47,48]. Data are provided by the game to assess motor performance.

The VR game captures the movements of the participant through a Webcam and does not need physical contact for the task to be performed. Instead, the participant moves their arms, at a one-meter distance from the computer screen. Alternatively, the Touch Screen can be used, which requires contact with the computer screen, in the inside space of the spheres. The participant needs to wait for the spheres that are falling from above to begin to overlap one of the target circles. The game offers hit feedback through numbering (+1) that appears on the side of any sphere successfully reached while in the target. In addition, the total score is visible in the left upper corner of the screen, marking 10 points for every hit.

### 2.4. Physical Activity by Percentage of Heart Rate Reserve (HRR)

All participants wore a chest strap for measurement of heart rate (HR), logged using the Polar V800 measuring device (Polar Electro Oy, Kempele, Finland), for beat-to-beat heart rate capture, with the heart rate receiver placed next to it [49]. The data collected were resting HR (RHR) and maximum HR achieved during the VR activity (HRvr). We used the convenient formula based on the person’s age to estimate maximal heart rate (HRmax= 220 − age) [50]. The difference between HR max and the resting HR gives the heart rate reserve (HRR). To estimate the %HRR achieved during the VR activity and then estimate the intensity of the activity, we considered the following formula [51,52,53]: %HRR = (HRvr − rHR/HRR) × 100

### 2.5. Enjoyment Scale

An enjoyment scale using smiley faces (0 “not fun at all”, 1 “boring”, 2 “a little bit fun”, 3 “fun”, and 4 “great fun”) was applied after the end of every day of VR activity, to verify the participant’s level of enjoyment while performing the activity. This scale was developed by Jelsma et al. [54] to evaluate how people feel when interacting with the proposed non-immersive VR games and was previously used in other studies using different games [45,55,56].

### 2.6. Randomization 

Two days after the recruitment process screening and initial evaluation (by the same researcher—D0), a simple randomization was performed, using the web site randomization.com, by an independent researcher, who was not involved with the participants’ recruitment or their evaluation. Figure 2 shows the study design and all protocol phases. The children or adolescents who were eligible to participate in the research were distributed randomly into 2 opposite sequences, with a ratio of 1:1. They carried out a 10 session protocol, twice a week, for 12 min per session, with 5 sessions practicing on each interface. Participants were involved for a total of six weeks (pre-intervention assessment D0, followed by 10 sessions). The intervention groups were: Sequence A: Beginning the intervention by performing a virtual task using the Webcam interface, followed by a real task, using the Touch Screen interface.Sequence B: Beginning the intervention by performing a real task using the Touch Screen interface, followed by a virtual task, using the Webcam interface.

### 2.7. Data Analysis

For the independent variables, the Student’s T test was used to compare Sequence A and Sequence B. For the dependent variables, the MoveHero, error measures were considered, defined as the difference between the moment when the sphere reached the target (arrival time) and the time when the touch or gesture was recorded in milliseconds, the measures being: Absolute Error (AE), to demonstrate the accuracy of movement, and Variable Error (VE), to identify the precision of the movement [57,58]. The quantity of hits (balls hit on the target) during the game was also considered as a dependent variable. To determine the intensity of the activity, we estimated the %HRR during the VR activity on each day of the intervention. 

The dependent variables are presented as mean and standard deviation and were submitted to MANOVA with 2 (Sequence: A virtual-real and B real-virtual) by 2 (Practice: First and second) by 5 (Days of training in each practice: D1 to D5) with repeated measures in the last two factors. The Least Significant Difference (LSD) was used as the post-hoc test. In addition, for the categorical variables of the Enjoyment Scale, a chi-square test was applied. 

Finally, a regression analysis was performed to determine whether the independent variables (Age, Height, Weight, Body Mass Index (BMI), Intelligence Quotient (IQ), Childhood Autism Assessment Scale (CARS) and Inventory of Pediatric Disability Assessment (PEDI)) influenced performance improvement, with the dependent variable being the difference in the AE between the fifth day and the first day of both practices (ΔD5 − D1). The graph data are presented as mean and standard error. The partial Eta squared (ŋ_p_^2^) was reported to measure the effect size and interpreted as small (effect size > 0.01), medium (effect size > 0.06), or large (effect size > 0.14) [59]. The statistical package used was SPSS, version 26.0, and we considered significant *p*-values < 0.05.

## 3. Results

In total, 30 children and adolescents with ASD were invited to participate in the study, of whom six did not sign the informed consent form, so 24 male participants were included in the protocol. One participant was subsequently excluded for not understanding the task and one withdrew during the study, not completing the training days, ending with a sample of 22 participants, randomized into two Sequences according to the practice performed (starting on the Touch Screen-real or WebCam-virtual). All the participants reported that they had previous experience of commercial non-immersive VR games. Table 1 shows the mean values of independent variables, as well as their dispersion values, demonstrating that the Sequence groups were homogeneous.

### 3.1. Software MoveHero—VR Activity

MANOVA revealed a significant effect for Days (F12, 09 = 7.62; *p* = 0.002, ŋ_p_^2^ = 0.91; Wilks’ λ = 0.581) and interactions were found for Practice and Sequence (F3, 18 = 77.7; *p* < 0.001, ŋ_p_^2^ = 0.92; Wilks’ λ = 0.072) and Practice, Days and Sequence (F12, 09 = 9.41; *p* = 0.001, ŋ_p_^2^ = 0.92; Wilks’ λ = 0.343). Separate follow-up repeated measures (RM-ANOVAs) for AE, VE and quantity of hits are reported in the paragraphs below.

#### 3.1.1. Absolute Error/Accuracy—AE

Considering AE, ANOVA found no main effect for any of the factors, but interactions were found for Practice and Sequence (F1, 20 = 96.4; *p* < 0.001; ŋ_p_^2^ = 0.83) and for Practice, Days and Sequence (F4, 80 = 16.4; *p* < 0.001; ŋ_p_^2^ = 0.45). The post-hoc test showed worsening between the first practice in Sequence B (which started the intervention in the real interface—Touch Screen), while the participants who performed the inverse Sequence A (starting in the virtual interface—WebCam) did not present significant differences in AE.

Additionally, a Student t test was carried out between the interfaces in each Practice, to verify if there was an influence of the first practice on the result of the second practice. We chose to compare the same interfaces in the different sequences (WebCam/virtual: first practice in Sequence A versus second practice in Sequence B; Touch Screen/real: first practice in Sequence B versus second practice in Sequence A). For the Touch Screen interface (see important result Figure 3), a significant difference was found on day 1 (*p* = 0.044), meaning that in Sequence A, which initially performed the Practice on the virtual interface (WebCam), when participants performed the Practice on the real interface (Touch Screen) on the first day after crossover, they presented a lower AE (M = 135 ms) than in Sequence B (M = 180 ms), which did not perform previous practice on the virtual interface. In turn, for the WebCam interface (virtual), there was no difference, indicating that previous practice did not influence performance on this interface. The regression analysis did not show significance, demonstrating that the independent variables had no influence on the performance improvement.

#### 3.1.2. Variable Error/Precision—VE

Figure 4 presents the VE during the training days; ANOVA revealed a significant effect for Sequence (F4, 80 = 7.20; *p* = 0.014; ŋ_p_^2^ = 0.26) and for Days (F4, 80 = 8.16; *p* <0.001; ŋ_p_^2^ = 0.29), in addition to interactions for Practice and Sequence (F1, 20 = 192.8; *p* < 0.001; ŋ_p_^2^ = 0.90) and for Practice, Days and Sequence (F4, 80 = 12.7; *p* < 0.001; ŋ_p_^2^ = 0.38). The post-hoc test showed that Sequence A had a lower VE (M = 179 ms) than Sequence B (M = 223 ms), indicating better precision in Sequence A. Finally, there was improvement between the first (M = 219 ms) and fifth day (M = 195 ms) of the intervention in all participants only during the practice on the virtual interface (WebCam).

For VE the same comparison as in AE was made, to verify if there was an influence of the first practice on the result of the second practice (WebCam/virtual: first practice in Sequence A versus second practice in Sequence B; Touch Screen/real: first practice in Sequence B versus second practice in Sequence A). For the Touch Screen interface, a significant difference was found on days 1 (*p* = 0.007), 2 (*p* = 0.006) and 3 (*p* = 0.021). These results show that for Sequence A which had initially performed the Practice on the virtual interface (WebCam), when participants performed the Practice on the real interface (Touch Screen) on the first day after crossover, they presented a lower VE (M = 120 ms) than the same Practice in Sequence B (M = 157 ms); these differences were maintained on the next two days. For the WebCam interface, a significant difference was found between days 1 (*p* = 0.003) and 2 (*p* = 0.025). These results show that for Sequence B, which had initially performed the Practice on the real interface (Touch Screen), when participants performed the Practice on the virtual interface (WebCam) on the first day after crossover, they presented a higher VE (M = 331 ms) than the same Practice in Sequence A (M = 266 ms), these differences remained on the second day.

#### 3.1.3. Quantity of Hits

ANOVA revealed a significant effect for Sequence (F1, 20 = 3.75; *p* = 0.067; ŋ_p_^2^ = 0.15) and for Days (F4, 80 = 17.13; *p* < 0.001; ŋ_p_^2^ = 0.46) and an interaction for Practice, Days and Sequence (F4, 80 = 3.47; *p* = 0.016; ŋ_p_^2^ = 0.14). Post-hoc comparisons showed that both Sequences increased the quantity of hits from the first (M = 23.6) to the fifth day (M = 26.3) and Sequence A had a greater quantity of hits (M = 26.9) than Sequence B (M = 23.9). This result was maintained in both groups on all practicing days (Figure 5).

### 3.2. Percentage of Heart Rate Reserve (%HRR)

Considering the %HRR achieved during the VR activity, a main effect was found for Sequence (F1, 20 = 5.16; *p* = 0.034; ŋ_p_^2^ = 0.20), a marginal effect for Practice (F1, 20 = 3.65; *p* = 0.070; ŋ_p_^2^ = 0.15) and an interaction between Sequence, Practice and Days (F4, 80 = 2.94; *p* = 0.034; ŋ_p_^2^ = 0.12). This finding indicates that Sequence A had a greater %HRR (M = 28.3%) than Sequence B (M = 24.9%). Both groups had a higher %HRR in the second Practice. Post-hoc comparisons showed that Sequence A had the highest %HRR on 3 days; D2 (*p* = 0.021) and D4 (*p* = 0.006) from the first Practice and D3 (*p* = 0.040) from the second Practice (Figure 6).

### 3.3. Enjoyment Scale

No statistical differences were found between Sequences for the enjoyment scale. The majority pf participants reported “great fun” in both Sequences, as shown in Figure 7.

## 4. Discussion

The present study investigated the influence of task practice in a real (Touch Screen interface) and virtual environment (WebCam interface) in children and adolescents with ASD, in a longitudinal protocol of 10 interventions. Our findings reinforce that practice in a virtual environment is more difficult but promotes a different strategy, allowing performance improvement on transfer of motor performance to the real environment regarding accuracy and precision of movement. Additionally, all participants presented an increase in heart rate during the interventions, which represents a very light (<30% HRR) to light intensity (< 39% HRR) according to the American College of Sports Medicine guidance for prescribing exercise [60,61] and most of the participants reported high enjoyment in both sequences. This result partially agrees with our hypothesis and we will discuss our data below.

### 4.1. Improvement in Motor Performance and Transfer to the Subsequent Practice When Changing the Environment

Considering both accuracy and precision (absolute and variable error), the participants who performed Sequence A (starting in virtual environment) presented an improvement after changing to the real environment (transfer). When comparing the same interface in the inverse sequence (6th day on sequence A with 1st day on sequence B), representing performance transfer from the virtual to real environment, the results did not show transfer of performance. Considering precision (variable error), participants of both sequences presented improved performance only in the virtual environment; on the other hand, for accuracy (absolute error) both sequences presented worsening in the real environment. Lastly, although both sequences presented an increased quantity of hits during practices, sequence A (virtual-real) had a greater quantity of hits than sequence B (real-virtual). These results can be justified by three specific factors that we will examine below:

a. Virtual Reality Environment: VR can provide learning in a controlled and safe environment for people with ASD, improving adherence and motivation [62] and also possibly enhancing transfer to daily activities in real life [63,64,65,66]. Corroborating the results of our study, Herrero et al. [67] also identified that people with ASD, when performing a motor activity in a virtual environment, presented improved performance in a real task (simple reaction time task). According to Bölte et al. [68], virtual reality is a powerful tool for optimizing the transfer of skills acquired after computer-based training to the real environment. This happens because, in virtual tasks, the objective of the movement is abstract and directed to intangible objects that can directly influence performance and, since these tasks require motor adaptations, the individuals do not simply learn the specific movements that need to be executed during the training, but they also acquire a perceptual motor ability that can be generalized to other situations. In other words, they can transfer their new learned ability to a different situation from the one in which they previously trained [31,69,70]. Our results also agree with Moraes et al. [33], who found an important improvement in timing error, when the participants (after practice) transferred from the virtual to real environment in a cross-sectional protocol, which indicates that virtual environments may enhance learning of movement in this group, also showing that the practice of sequence repetition promotes a faster motor response. In a similar way, Saiano et al. [66] observed that training pedestrian abilities in a virtual environment led to better improvement in a transference to real life behaviors and promoted better involvement and motivation. Participants performed worse in the virtual environment when compared to the real environment. This can be justified by the speed–accuracy trade-off, which proposes that the more difficult the task, the longer the time needed to complete it [71,72,73,74].

b. Feedback: People with ASD have a long-recognized difficulty in sensory (including tactile) processing when compared to typically developed children. Sensorimotor representation can have a role in the motor function when the individual is learning new movements [75]. We hypothesize that the multiple feedback given by the MoveHero game leads to better performance, considering that the visual, auditory and somatosensory systems are important in maintaining attention and, as an adaptive guide to virtual environments, the game provides visual and auditory feedback when the players miss the ball or hit them correctly. These multiple feedback options have been shown to be quite useful for providing an adaptive benefit and, when combined, they seem to promote a faster response to a stimulus, when compared to single sensorial feedback in people with ASD [76,77]. Moreover, Gidley Larson and Mostofsky [10], Haswell et al. [78], Izawa et al. [79], Marko et al. [11] and Sharer et al. [75] suggested the importance of learning and the necessity for proprioceptive errors to finish a task in children with ASD. The lower tactile feedback [80] and the movement demand from the virtual environment possibly also increased the level of difficulty of the task. Thus, as we hypothesized, the task in the virtual environment seems to be more difficult; however, all these characteristics of the non-contact environment promoted better transfer of the task, as discussed previously.

c. Music: Although the stimulus from music was not one of our goals, the MoveHero game plays a song throughout the practice, which may have contributed to the increased performance during the virtual task in people with ASD. There is growing evidence that music can be of strong benefit to people with ASD, because it can activate neural circuits of reward in this population [81,82]. Some current researchers found that the practice of listening to music can possibly improve attention in people with ASD and could also be a sign of the improvement in sensory integration and sensorimotor functioning [83]. In a recent review by Quintin et al. [82] the authors concluded that music is a powerful therapeutic tool for people with ASD and is a field to explore, because it promotes many benefits, such as activation of neural circuitry associated with emotions, which could be promising and helpful in increasing understanding about ASD deficits. Sharda et al. [84] reported the first evidence that musical interventions of 8–12 weeks increase the connectivity between the bilateral primary auditory cortex and subcortical and motor regions, which are regions usually affected in ASD, and improvements in the children’s communication were reported after this intervention.

### 4.2. Promote Physical Activity by Altering the Heart Rate

Considering physical activity, Sequence A (virtual-real) had a greater % of heart rate reserve (%HRR) than Sequence B (real-virtual), but both groups had a higher %HRR in the second practice, showing the long-term benefit of the proposed activity. MoveHero is a game that can promote many benefits, such as motor learning improvement and an increase in levels of physical activity [43]. Considering the %HRR achieved during the VR activity, we can affirm that the activity in both Sequences had a very light (<30% HRR) to light intensity (<39% HRR) [60,61].

In this way, a recent systematic review aimed to examine the effects of VR based exercise on different outcomes and in various populations. The authors suggest that VR exercise has the potential to exert a positive impact on an individual’s physiological, psychological and rehabilitative outcomes compared with traditional exercise [85]. In turn Eichhorn et al. [86] using an upper limb movement task similar to our study, carried out a pilot study in healthy students, attempting to obtain changes in the heart rate through movement of the upper limbs. The target heart rate for individual users was achieved, confirming the potential of VR regarding entertaining aspects and physical effort. Naugle et al. [87] found similar results using Wii games in 10 min sessions, at a self-selected intensity that ranged from “very light” when playing Wii tennis to “somewhat hard” when playing Wii boxing. This appears to be insufficient activity for maintaining or improving cardiovascular fitness. Our task lasted only 12 min and we observed only a slight increase in heart rate, so we can suggest that a longer intervention time, with a greater range of motion, could perhaps generate changes compatible with moderate or intense physical activity, demonstrating the potential of activity in virtual reality.

### 4.3. Enjoyment

In both sequences the majority of the enjoyment scores remained in the “fun” and “great fun” levels on all the intervention days. Although there was no difference between the sequences, we observed that, in sequence A (virtual-real), the “a little bit fun” score appeared only once, so we can speculate that maintaining the level of enjoyment in the group in which the intervention was initiated by the virtual environment may have been one of the factors that contributed to the good performance. On the other hand, this score appeared on the first day of the intervention in the real environment, which may show that the real task was not as interesting for the participants; this was also the task in which a worsening in motor performance occurred. A recent review by Wang et al. [88] showed that motivation is one of the main factors that promotes better performance in VR therapy.

Promoting a fun and playful environment for the ASD population is quite a challenge, as they have many fixed interests and may have difficulty engaging in pretend play, so they can get angry or frustrated with some types of games or activities [89]. When these children do not feel any restrictions on their play, it tends to be more effective [90]. It is possible that, if we had a more flexible protocol in which they could choose the game, or the music, engagement may be higher. The results of many studies have shown that enjoyment is a significant predictor of PA participation regardless of the age of the participants [28]. Debska et al. [23] found a high enjoyment rating during PA in VR. Considering daily PA participation in adolescents with disabilities, Jin et al. [91] found that those who enjoyed PA participated more and those who spent more days being physically active were healthier than their counterparts. It was noted that, to promote health activities, the PA needs to be enjoyable.

Finally, we suggest that rehabilitation teams can organize intervention programs using technology and virtual environments to enhance learning in people with ASD, facilitating the transfer of motor performance to the real environment, while maintaining enjoyment and generating light physical activity in a controlled and safe environment. 

The limitations of this study include the fact that we did not include heart rate variability analysis that could have provided more extensive data [92,93,94]. In addition, we did not include female participants, so we cannot generalize our data to this population. Studies with a large female group could yield important results. The same is true for participants with severe autism, who may not have understood the task. It may be the case that in considering more severe ASD use of a virtual environment may be less useful for this population; they may not be able to perform the task. Another limitation can be identified from considering the music used during practice; the researcher randomly chose existing music in the MoveHero software. Greater selection and control of music selection could offer further trends in the results and should be considered in future work. We also suggest that future studies would benefit from incorporating, a longer period of time for the activity and a wider range of movements or more intense physical activity.

## 5. Conclusions

The virtual reality activity used in our study had a higher level of difficulty, but provided transference to the real environment. The participants with ASD presented improvements in performance, mainly in the virtual environment. In terms of physical activity, our task provided very light to light physical activity assessed by changing heart rate and the majority of participants enjoyed the task.

## Figures and Tables

**Figure 1 ijerph-19-14668-f001:**
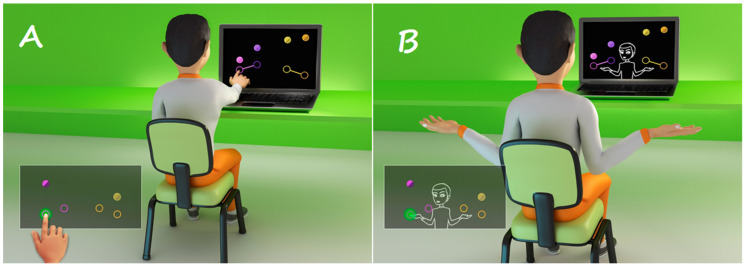
Representative design of a participant playing the MoveHero software task (Research group and technological application in rehabilitation—PATER group, São Paulo, Brazil), with the use of the (**A**) Touch Screen interface—real and (**B**) WebCam interface—virtual.

**Figure 2 ijerph-19-14668-f002:**
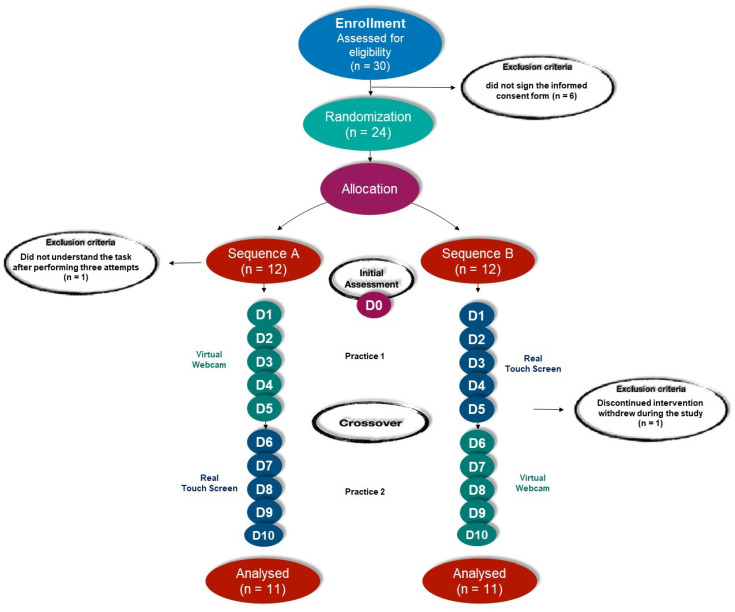
Study design (D1–D5: Days of intervention).

**Figure 3 ijerph-19-14668-f003:**
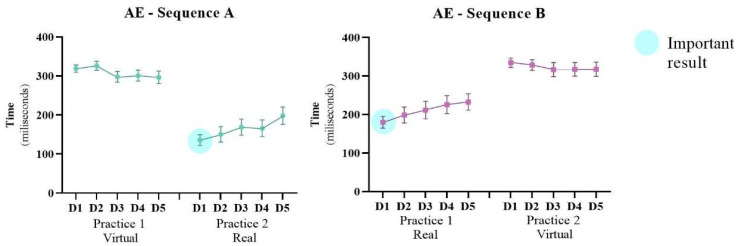
Representation of mean and standard error of the Absolute Error (AE) of both Sequences (A: virtual → real and B: real → virtual) during the 10 days of intervention (D1–D5: Days of intervention). ● Important result: The participants that practiced first in the virtual interface (Sequence A) had better performance in the second practice with the real interface and this sequence presented better results than the inverse (Sequence B). This can be reinforced by the statistical difference in comparison between the first day after crossover in Sequence A with the first day of the first practice in Sequence B (marked with the blue sphere).

**Figure 4 ijerph-19-14668-f004:**
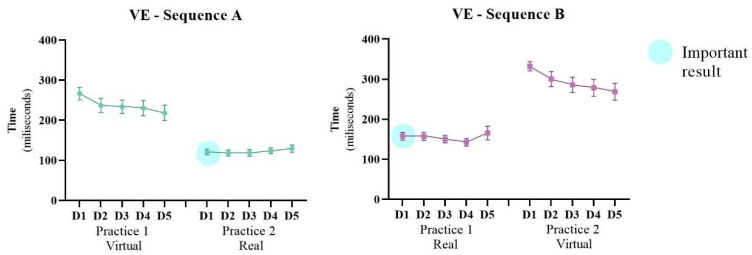
Representation of mean and standard error of the Variable Error (VE) of both Sequences (A: virtual → real and B: real → virtual) during the 10 days of intervention (D1–D5: Days of intervention). ● Important result: The participants that practiced first in the virtual interface (Sequence A) had better performance in the second practice with the real interface and this sequence presented better results than the inverse (Sequence B). This can be reinforced by the statistical difference in comparison between the first day after crossover in Sequence A with the first day of the first practice in Sequence B (marked with the blue sphere).

**Figure 5 ijerph-19-14668-f005:**
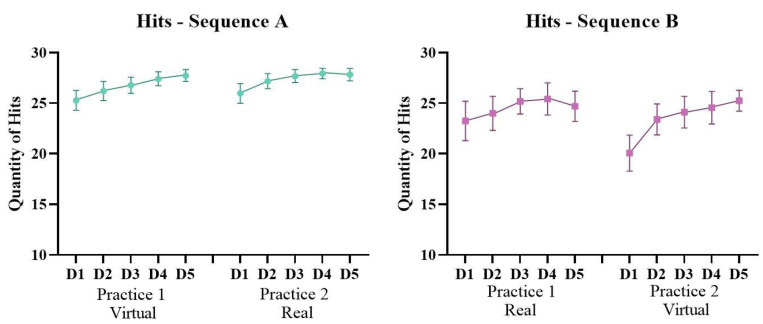
Representation of mean and standard error of the quantity of Hits of both Sequences (A: virtual → real and B: real → virtual) during the 10 days of intervention (D1–D5: Days of intervention).

**Figure 6 ijerph-19-14668-f006:**
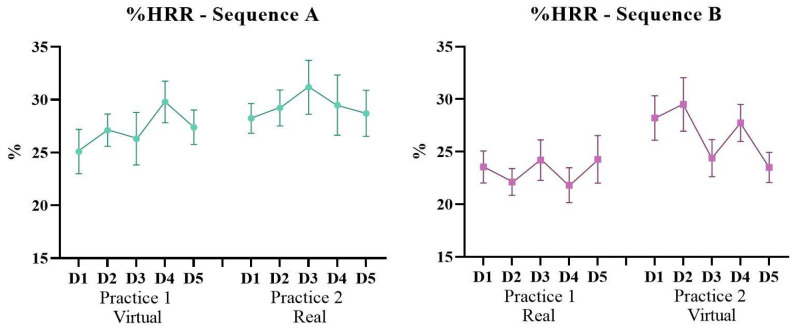
Graphic representation of the %Heart Rate Reserve (%HRR) achieved during the VR activity of Sequences A (virtual–real) and B (real–virtual) during the 10 days of intervention (D1–D5: Days of intervention).

**Figure 7 ijerph-19-14668-f007:**
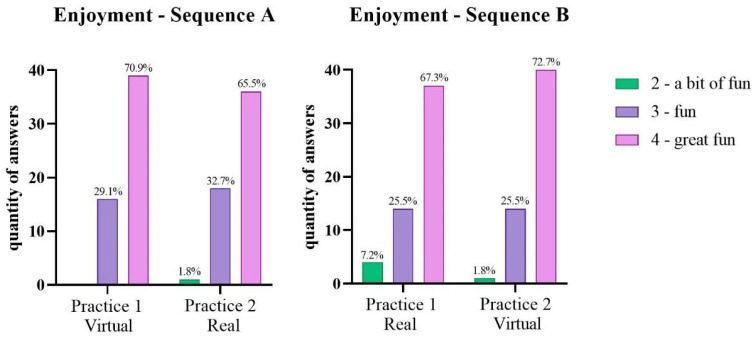
Graphic representation of the frequencies of each enjoyment score in Sequences A (virtual–real) and B (real–virtual).

**Table 1 ijerph-19-14668-t001:** Comparison of the characterization variables between Sequences A (virtual–real) and B (real–virtual).

	Sequence A	Sequence B	*p*-Value
	Mean ± SD	Mean ± SD	
	CI (LL, UL)	CI (LL, UL)	
Age—years	14.1 ± 1.7	13.9 ± 1.9	0.822
	(12, 15)	(12, 15)	
Height—meters	1.65 ± 0.13	1.67 ± 0.11	0.693
	(1.57, 1.73)	(1.59, 1.75)	
Weight—kilograms	61.9 ± 14.8	63.0 ± 18.8	0.877
	(51, 72)	(52, 73)	
BMI	22.0 ± 4.0	22.0 ± 6.0	0.899
	(19, 25)	(19, 25)	
IQ	87.6 ± 11.3	83.13 ± 13.6	0.412
	(79, 95)	(75, 91)	
CARS	34.3 ± 1.3	33.7 ± 1.7	0.348
	(33, 35)	(32, 34)	
PEDI—FS	89.6 ± 11.3	91.8 ± 10.1	0.648
Self-Care	(82, 96)	(85, 98)	
PEDI—FS	66.3 ± 7.7	67.7 ± 1.6	0.586
Mobility	(62, 69)	(64, 71)	
PEDI—FS	77.5 ± 9.8	77.8 ± 9.9	0.947
Social Function	(71, 83)	(71, 84)	
PEDI—CA	94.2 ± 9.0	97.0 ± 6.7	0.419
Self-Care	(89, 99)	(92, 102)	
PEDI—CA	92.4 ± 15.0	96.8 ± 5.7	0.377
Mobility	(85, 99)	(89, 103)	
PEDI—CA	88.0 ± 11.5	87.1 ± 12.8	0.867
Social Function	(80, 95)	(79, 94)	

SD: Standard Deviation; CI: Confidence Interval; LL: Lower Limit; UL: Upper limit; BMI: Body Mass Index; IQ: Intelligence Quotient; CARS: Childhood Autism Rating Scale; PEDI: Pediatric Disability Assessment Inventory; FS: Functional Skills; CA Caregiver Assistance.

## Data Availability

The data presented in this study are available on request from the corresponding author. The data are not publicly available due to other studies in progress.

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
