# Peer review of "Effect of Longitudinal Practice in Real and Virtual Environments on Motor Performance, Physical Activity and Enjoyment in People with Autism Spectrum Disorder: A Prospective Randomized Crossover Controlled Trial"

_ijerph, 2022, doi:10.3390/ijerph192214668_

Round 1
Reviewer 1 Report
The present study is well structured and can be extremely useful for professionals working with children with ASD, covering a sphere of abilities of these children that is not much studied (as opposed to the sphere of language or social behavior).
The description of the experiment itself is very clear and the attention to detail is to be appreciated.
The statistical analyzes and the results obtained are comprehensive and presented in a logical and well-organized manner.
The longitudinal design is an important strength of the study.
We suggest that the first sentence from 2.1 Participants (lines 101-104) be reformulated, it is difficult to follow.
Line 107 – assent term must be the informed consent, maybe it would be better to replace the first sentence with the second
Line 112 – after word `blockers' must be a dot
Lines 106-115 – the inclusion/exclusion criteria are explained too complicated... for example, it seems more appropriate that the inclusion criterion "not having associated disorders" can be an exclusion criterion ("having associated disorders")
Line 211 – an inexplicable 2 appears... must be explained
Line 231 – 30 ASD sounds very impersonal... maybe 30 children with ASD
Line 258 – what kind of analysis?
There is no "limitation" section declared by authors.
Reviewer 2 Report
The manuscript entitled “Effect of longitudinal practice in real and virtual environments on motor performance, physical activity, and enjoyment in people with Autism Spectrum Disorder: a prospective randomized crossover controlled trial” is focusing on an interesting topic related to the impact of activity in virtual and real environments on improvement in motor performance physical activity and enjoyment in young people with ASD.
The overall quality of this work is good and meets the basic requirements of the journal. The results, methods, and data interpretations were clear. Literature searching and citation papers support the main content.
Minor:
-
in the abstract Authors stated that “The participants 31 performed a 10 session protocol, with 5 sessions practicing in each VR environment (virtual or real)” - VR is always virtual, so it can not be real, please revise throughout the text;
-
what basis the song was chosen? if this song was correlated with the preferences and hearing of participants?;
-
please define: a very light to light intensity;
-
in my opinion, the Authors should explain why this particular game was chosen for people with ASD;
-
please change the position of the figure description, it should be at the bottom ;
-
please include the reference Matsangidou et al., 2019 which is cited in the Introduction but not listed in the references list.
